# Establishment of the Embryonic Shoot Meristem Involves Activation of Two Classes of Genes with Opposing Functions for Meristem Activities

**DOI:** 10.3390/ijms21165864

**Published:** 2020-08-15

**Authors:** Mitsuhiro Aida, Yuka Tsubakimoto, Satoko Shimizu, Hiroyuki Ogisu, Masako Kamiya, Ryosuke Iwamoto, Seiji Takeda, Md Rezaul Karim, Masaharu Mizutani, Michael Lenhard, Masao Tasaka

**Affiliations:** 1International Research Organization for Advanced Science and Technology (IROAST), Kumamoto University, 2-39-1 Kurokami, Chuo-ku, Kumamoto 860-8555, Japan; 2Graduate School of Biological Sciences, Nara Institute of Science and Technology, 8916-5 Takayama, Nara 630-0192, Japan; moritsubaki.panda@gmail.com (Y.T.); s-s.combination@docomo.ne.jp (S.S.); h.ogisu@icloud.com (H.O.); masadecosmos@gmail.com (M.K.); r.ftvm22@ymobile.ne.jp (R.I.); seijitakeda@kpu.ac.jp (S.T.); mrkarim1996@yahoo.com (M.R.K.); m-tasaka@bs.naist.jp (M.T.); 3Graduate School of Life and Environmental Sciences, Kyoto Prefectural University, Hangi-cho 1-5, Shimogamo, Sakyo-ku, Kyoto 606-8522, Japan; 4Department of Horticulture, Bangladesh Agricultural University, Mymensingh 2202, Bangladesh; 5Graduate School of Agricultural Science, Kobe University, 1-1 Rokkodai, Nada-ku, Kobe 657-8501, Japan; mizutani@gold.kobe-u.ac.jp; 6University of Potsdam, Institute for Biochemistry and Biology, Karl-Liebknecht-Str. 25-26, 14476 Potsdam-Golm, Germany; michael.lenhard@uni-potsdam.de

**Keywords:** shoot meristem, embryogenesis, stem cell, boundary, transcription factor, cytochrome P450, *CUC*, *STM*, *LAS*, *BLR*, *KNAT6*, *KLU*, *CYP78A5*

## Abstract

The shoot meristem, a stem-cell-containing tissue initiated during plant embryogenesis, is responsible for continuous shoot organ production in postembryonic development. Although key regulatory factors including *KNOX* genes are responsible for stem cell maintenance in the shoot meristem, how the onset of such factors is regulated during embryogenesis is elusive. Here, we present evidence that the two *KNOX* genes *STM* and *KNAT6* together with the two other regulatory genes *BLR* and *LAS* are functionally important downstream genes of *CUC1* and *CUC2*, which are a redundant pair of genes that specify the embryonic shoot organ boundary. Combined expression of *STM* with any of *KNAT6*, *BLR*, and *LAS* can efficiently rescue the defects of shoot meristem formation and/or separation of cotyledons in *cuc1*
*cuc2* double mutants. In addition, *CUC1* and *CUC2* are also required for the activation of *KLU*, a cytochrome P450-encoding gene known to restrict organ production, and *KLU* counteracts *STM* in the promotion of meristem activity, providing a possible balancing mechanism for shoot meristem maintenance. Together, these results establish the roles for *CUC1* and *CUC2* in coordinating the activation of two classes of genes with opposite effects on shoot meristem activity.

## 1. Introduction

Primary growth in plant shoots depends on the activity of stem-cell-containing tissue called the shoot meristem, which is located at the tip of the stem [1]. The shoot meristem is initially formed during embryogenesis and is activated upon germination to produce shoot organs such as leaves, stems, and floral organs, while it maintains an undifferentiated stem cell population at its center. Once activated, the shoot meristem keeps the balance between cell proliferation and differentiation to maintain an appropriate size of the stem cell population within it; factors essential for this process have been identified [2,3]. Although activation of these maintenance factors is associated with shoot meristem initiation during embryogenesis, how the process is coordinated is unknown.

Several key regulators for shoot meristem initiation have been reported [4,5,6,7,8,9,10,11]. Among them, the NAM/CUC3 type of NAC-domain transcription factors represents a class of regulators that are required for specification of shoot organ boundaries, which are sites for shoot meristem formation in embryonic and postembryonic development [12,13,14,15,16]. In *Arabidopsis thaliana*, the three NAM/CUC3 genes *CUC1*, *CUC2*, and *CUC3* are expressed in cells along the boundary between two cotyledon primordia and promote shoot meristem formation and the separation of cotyledons [5,6,17]. As shoot meristem formation proceeds, expression of these genes is downregulated from the meristem center and becomes restricted to the adaxial and lateral boundaries of cotyledons. In postembryonic development, the three *CUC* genes are expressed at the adaxial and lateral boundaries of leaf primordia and are required for the formation of axillary shoot meristem as well as for the separation of leaves [6,18].

Several genes whose expression is dependent on *CUC* gene activities have been identified. Expression of the two *KNOTTED1-like homeobox* (*KNOX*) genes *SHOOT MERISTEMLESS* (*STM*) and *KNOTTED1-like from Arabidopsis thaliana 6* (*KNAT6*), which are required for shoot meristem maintenance, is absent from the *cuc1 cuc2* double mutant [17,19] and ectopic expression of the *CUC* genes induces *STM* expression [5,20,21]. The *LIGHT-DEPENDENT SHORT HYPOCOTYLS* (*LSH*) genes *LSH3* and *LSH4*, which encode nuclear proteins of the *Arabidopsis* LSH1 and *Oryza* G1 (ALOG) family, have been identified as direct transcriptional targets of the CUC1 protein and their overexpression induces ectopic shoot meristem formation [22]. Genome-wide mapping of protein–DNA interactions among boundary-enriched genes has identified the *GRAS* family gene *LATERAL SUPPRESSOR* (*LAS*) and the microRNA gene *MIR164C* as direct transcriptional targets of CUC2 [23]. *LAS* encodes a putative transcriptional regulator and is required for axillary shoot meristem formation [24]. Together, these analyses indicate that *CUC* genes regulate multiple genes involved in shoot meristem activity or boundary specification, or both. However, the functional relationship between these downstream genes and *CUC* gene activity remains elusive.

Here, we selected a set of genes whose expression is dependent on *CUC1* and *CUC2* during embryogenesis and demonstrated that the combined activities of *STM*, *KNAT6*, *BLR*, and *LAS* are important for promoting shoot meristem formation and cotyledon separation downstream of *CUC1* and *CUC2*. Moreover, *CUC1* and *CUC2* are also required for the expression of *KLUH* (*KLU*)/*CYP78A5*, a cytochrome P450-encoding gene involved in the rate of shoot organ production and organ size [25,26]. Genetic analysis indicates that *KLU* restricts shoot meristem activity and counteracts *STM* function. Our results thus indicate that the activation of two classes of genes with opposing functions, one positively and the other negatively affecting meristem activity, is an important step for shoot meristem formation.

## 2. Results

### 2.1. Selection of Candidate CUC1 and CUC2 Downstream Genes

It has been reported that the two *KNOX* genes *STM* and *KNAT6*, the GRAS gene *LAS*, and the two *ALOG* genes *LSH3* and *LSH4* show overlapping expression patterns to those of *CUC1* and *CUC2* in the boundary region of cotyledons, and their expression is absent in the corresponding region of *cuc1 cuc2* double-mutant embryos [5,6,17,19,22]. We identified six additional candidate downstream genes positively regulated by *CUC1* and/or *CUC2* from microarray-based screening combined with quantitative real-time polymerase chain reaction (qRT-PCR) and in situ hybridization experiments (Appendix A). These genes were expressed in the boundary region that overlapped with the *CUC* gene expression domain [5,17] and were downregulated specifically in the corresponding region of *cuc1 cuc2* embryos (Figure 1A–D; Appendix A), indicating the dependence of their expression on *CUC1* and *CUC2* activities during embryogenesis.

To gain insight into how CUC1 regulates expression of these candidate genes, we used the glucocorticoid receptor (GR) system, in which the activity of CUC1 is induced by the exogenous application of dexamethasone (DEX) [27]. Using this system, we previously found evidence for the direct activation of the *LSH3*, *LSH4*, and *STM* genes by CUC1 [22,28]. Among the remaining eight genes, we found that only *LAS* and *PAN* were significantly upregulated upon DEX treatment alone in *CUC1-GR* plants (Figure 1E, left panel). By contrast, treating with both DEX and the protein synthesis inhibitor cycloheximide (CHX), which blocks secondary transcriptional responses caused by genes directly activated by CUC1, significantly upregulated not only *LAS* and *PAN*, but also four additional genes among the eight tested. Treatment with CHX alone did not alter their expression levels (Figure 1E, right panel). Together, the results suggest that the six genes are under direct transcriptional regulation by the CUC1-GR protein, but that the action of CUC1 is counteracted by a CHX-sensitive negative factor with respect to activation of four of the genes (*KLUH*, *KNAT6*, *UFO*, and *SAI-LLP1*). Another possibility is that DEX treatment alone indirectly promotes expression of genes that negatively affect CUC1-dependant activation of the four genes. In control non-transgenic plants, which do not express CUC1-GR, none of the genes were upregulated by DEX treatment in the presence or absence of CHX (Figure 1F), indicating that the induction of the downstream genes was not a secondary effect of DEX or CHX.

### 2.2. Combined Expression of STM with LAS, BLR, and KNAT6 is Sufficient to Rescue the Embryonic Shoot Phenotypes of cuc1 cuc2

To examine the functional significance of the candidate genes in processes downstream of *CUC1* and *CUC2*, we expressed each gene in the *cuc1 cuc2* double-mutant background under the control of the *CUC2* promoter (*ProCUC2*)*,* which drives expression in the boundary region in both wild-type and double-mutant embryos (Figure 2A,B). In wild type, seedlings develop a shoot that continuously produces leaves immediately after germination and have two completely separated cotyledons (Figure 2C). By contrast, the *cuc1 cuc2* double mutant has two cotyledons fused along their margins and fails to form a shoot, and this phenotype is fully penetrant (Figure 2D) [4]. When the coding sequence of *CUC2* was used as a positive control (*ProCUC2:CUC2*), 41.7% of the T1 seedlings showed a strongly rescued phenotype with no or slight delay in shoot formation and with completely separated cotyledons, 50.0% showed a mildly rescued phenotype with delayed or no shoot formation and half-separated cotyledons, and the remaining 8.3% showed non-rescued phenotype identical to that of *cuc1 cuc2* (Table 1, Appendix A).

Among the 10 downstream genes, the *ProCUC2:STM* and *ProCUC2:LAS* transgenes were able to mildly rescue the *cuc1 cuc2* phenotype, resulting in the occasional formation of a functional shoot that can produce leaves as well as in the partial separation of cotyledons (Table 1, Figure 2E, and Appendix A). In cleared seedlings, wild type has the dome-shaped shoot meristem with a few leaf primordia, whereas *cuc1 cuc2* plants lack either structure (Figure 2F,G) [4]. On the other hand, plants partially rescued by *ProCUC2*:*STM* showed variable phenotypes: some lacked a shoot meristem and leaf primordia, some developed small undifferentiated tissue, and the other produced the shoot meristem and leaf primordia (Figure 2H). These results indicate that *STM* and *LAS* play prominent roles in shoot meristem formation and cotyledon separation and that their individual activities can partially bypass the requirements for *CUC1* and *CUC2* for embryonic shoot meristem formation and cotyledon separation.

The rescue of the *cuc1 cuc2* mutant phenotype by the *STM* or *LAS* transgene alone was only mild and partial, thus we next tested their combined activities. In the F_2_ generation of the cross between the lines with the *STM* and *LAS* transgenes, only plants carrying both showed a rescued phenotype (Table 2). These rescued plants showed either partial (Figure 2J) or complete separation of cotyledons (Figure 2K), with the latter forming leaves with only a slight delay compared with the timing in the wild type, indicating that combined expression of *STM* and *LAS* is sufficient to compensate for the loss of *CUC1* and *CUC2* activities.

We next selected five other genes encoding transcription factors or transcriptional co-regulators, and tested their ability to rescue the *cuc1 cuc2* phenotype in combination with the *STM* transgene (Table 3). *BLR* and *KNAT6* were able to achieve rescue when combined with *STM* (Figure 2K,L), while the rest failed to do so. Plants expressing both *STM* and *BLR* produced nearly normal shoots with completely separated cotyledons (Figure 2K), indicating that, similarly to *LAS*, *BLR* can efficiently support the ability of *STM* to promote shoot meristem formation and cotyledon separation in the absence of *CUC1* and *CUC2*. By contrast, plants expressing both *STM* and *KNAT6* only rescued the cotyledon phenotype, but not that of shoot formation (Figure 2L), indicating that *KNAT6* can support the *STM* activity only in the limited developmental pathway downstream of the *CUC* genes.

### 2.3. Combined Loss of Function of STM, LAS, BLR, and KNAT6 Severely Impairs Shoot Meristem Formation and Cotyledon Separation

We then tested the combined effect of the loss-of-function mutations in the four genes (*STM*, *LAS*, *BLR*, and *KNAT6*) in young seedlings. Among the single mutants of these genes, only *stm* shows defects in shoot development with complete penetrance. In the case of the strong allele *stm-1C* [29], the mutant typically stops leaf formation after producing the first two leaves (Figure 3A–D). It has been reported that mutations in *BLR* or *KNAT6* alone do not cause visible phenotypes in seedlings, but enhance the defects of *stm* mutants [19,30]. Consistent with this, we found that the *stm-1C blr* double mutant produced fewer leaves than *stm-1C* (Figure 3A,C,D) and showed delayed first-leaf growth (Figure 3C,E). In addition, *stm-1C blr* showed a higher frequency of cotyledon fusion than *stm-1C* (Table 4). The *stm-1C knat6* double mutant showed strong cotyledon fusion and the lack of leaf production and both phenotypes were observed with complete penetrance (Figure 3F). These results confirmed the previous results that *BLR* and *KNAT6* are required for shoot meristem formation and cotyledon separation.

Similar to *blr* and *knat6* single mutants, young seedlings of the *las* single mutant reportedly show a normal appearance, except for a very small proportion of plants with fused cotyledons (0.26%) [6]. However, we found that the *las* mutation enhanced defects in shoot meristem activity when combined with the *stm-1C* single or *stm-1C blr* double mutant (Figure 3A–E). In addition, the *las* mutation enhanced the cotyledon fusion phenotype of both *stm-1C knat6* and *stm-1C blr knat6* mutants (Figure 3F–H). These results show that the *LAS* gene contributes to embryonic shoot meristem formation and cotyledon separation independently of *STM*, *KNAT6*, and *BLR*. The *blr knat6 las* triple mutant was phenotypically normal. Together, our results demonstrate that the four transcription factor-encoding genes *STM*, *KNAT6*, *BLR*, and *LAS* play key roles for embryonic shoot meristem formation and cotyledon separation downstream of CUC1 and CUC2.

### 2.4. The KLUH Gene Restricts the Embryonic Shoot Meristem and Counteracts STM

Among the downstream target genes, *KLU*/*CYP78A5* encoding a cytochrome P450 enzyme of the *CYP78A* family plays postembryonic roles in shoot organ size and organ production rate [25,26], raising the possibility that this gene also affects shoot meristem activity. Moreover, the mutation in rice *PLA1*, a member of the same family, causes an enlarged shoot meristem phenotype [31,32]. Indeed, we found that the two independent *klu* insertion alleles (*klu-019348* and *klu-4*) were associated with precocious leaf initiation in young seedlings (Figure 4A–F), suggesting enhanced shoot meristem activity. In addition, the width of the shoot meristem was significantly greater in *klu* than in the wild type (Figure 4G). Furthermore, embryos of the *klu* mutants displayed an enlarged cotyledon boundary region (Figure 4H–J) and this phenotype was associated with an enlarged expression domain of the shoot stem cell marker *CLV3::GUS* [33] (Figure 4K,L). These results indicate that the *KLU* gene is required for restricting shoot meristem size and activity during embryogenesis and postembryonic development.

The identification of *KLU* as a downstream gene of *CUC1* and *CUC2* was unexpected because *KLU* negatively affects shoot meristem activity, whereas the *CUC* genes are positive regulators of shoot meristem formation. To further investigate the relationship between *KLU* and the *CUC* genes, we crossed the *klu* mutant with the *cuc1 cuc2* double mutant and examined their genetic interactions. In single mutants of *klu*, *cuc1*, and *cuc2*, all seedlings produced a functional shoot as the wild type, and except for a small fraction of *cuc1* mutants with partially fused cotyledons, their cotyledons were completely separated (Table 5; Figure 5A). Similarly, seedlings of *klu cuc1* and *klu cuc2* double mutants all produced a fully functional shoot and most of them developed completely separated cotyledons, whereas small fractions had partially fused cotyledons (Table 5; Figure 5B). Importantly, the *klu cuc1 cuc2* triple mutant was indistinguishable from *cuc1 cuc2* double mutants in that they formed strongly fused cup-shaped cotyledons and lacked a shoot meristem (Table 5; Figure 2D,G and Figure 5C–E). These results show that the *cuc1 cuc2* double mutations are epistatic to *klu* and indicate that the *KLU* gene can affect shoot meristem activity only when the functional *CUC1* and *CUC2* genes are present.

Next, we examined the genetic interaction of *KLU* with *STM*, which represents a functionally important class of *CUC* downstream genes that positively affects shoot meristem activity. Young seedlings of the *stm klu* double mutant produced more leaves and had an enlarged shoot meristem compared with *stm* single mutants (Figure 5F–K). Moreover, whereas the strong *stm-1C* mutant allele typically arrested shoot growth after producing a few leaves (Figure 5L), the *stm-1C klu-4* double mutant showed prolonged vegetative shoot growth (Figure 5M). Taken together, these results indicate that the *KLU* gene activity counteracts that of *STM* in postembryonic shoot development.

To clarify the mechanism by which *CUC1* and *CUC2* regulate *KLU* gene expression, we examined the expression of reporter genes containing *cis*-regulatory sequences of *KLU*. A reporter construct that carried regions 2 kb upstream and 0.6 kb downstream (*Pro2kb*) showed activities in the cotyledon boundary region, cotyledon margins, and root pole (Figure 6A,B). This expression pattern resembled that of *KLU* mRNA detected by in situ hybridization, except that the reporter activity in cotyledons was broader and that in the root pole was detected for a prolonged time (compare Figure 6B with Figure 1A and Appendix A), indicating that the *Pro2kb* reporter contained a set of *cis*-regulatory elements sufficient for driving the native expression pattern of the gene at least in the cotyledon boundary region. When this construct was introduced into the *cuc1 cuc2* double-mutant background, its expression disappeared specifically in the cotyledon boundary region (Figure 6C), which corresponds to the region where the *CUC* genes act in normal development. Moreover, deletion of the region between 1 and 2 kb upstream of the gene resulted in the loss of expression in the cotyledon boundary region as well as in cotyledon margins (Figure 6A,D; *Pro1kb*). These results indicate that this 1 kb region contains *cis*-regulatory elements required for CUC1- and CUC2-dependent transcription in the shoot apex.

## 3. Discussion

In this work, through functional analyses of genes acting downstream of *CUC1* and *CUC2*, we found that the combined activities of *STM* with *LAS*, *BLR*, and *KNAT6* were important for shoot meristem formation and cotyledon separation. The abilities of these genes to rescue the *cuc1 cuc2* mutant phenotypes when expressed under the boundary-specific promoter, together with the strong shoot meristem and cotyledon phenotypes observed in the quadruple mutant, support a model in which the *CUC* genes promote shoot meristem formation and cotyledon separation mainly through the activation of these four genes. It was previously shown that *STM* is a direct transcriptional target of *CUC1* [28]. The experiments using the DEX-inducible CUC1-GR plants and CHX treatments in our current work indicate that *LAS* and *KNAT6* are additional direct targets of the CUC1 protein, whereas the regulation of *BLR* by CUC1 may be indirect.

Our analysis highlights the importance of the activation of *STM* expression by the *CUC* genes in shoot meristem formation and cotyledon separation. *STM* encodes a KNOX transcription factor [34] whose activity is continuously required for shoot meristem maintenance in postembryonic development through regulating various aspects of shoot meristem properties including pluripotency, self-maintenance, promotion of cell cycle, repression of differentiation, and hormone metabolism [28,35,36,37]. How *STM* promotes cotyledon separation is currently unknown, but its ability to repress growth and promote leaf dissection, when ectopically expressed, may be involved in the repression of growth at the cotyledon boundary [38].

The possible molecular mechanisms by which *STM* acts in concert with the rest of the four genes may vary. *KNAT6* encodes a KNOX protein closely related to STM, so its ability to enhance the rescuing activity of *STM* can simply be explained by functional redundancy [19,39]. By contrast, *BLR* encodes a BEL-class homeodomain protein, which physically interacts with STM [30], and nuclear localization of STM requires its interaction with BLR [40,41,42]. These results indicate that the coexpression of *BLR* with *STM* provides a sufficient amount of BLR–STM complex to the nucleus, thereby efficiently promoting shoot meristem formation and cotyledon separation. In axillary shoot meristem formation, another BEL protein, ATH1, which is functionally redundant with BLR, forms a heterodimer with STM and directly promotes the transcription of *STM*, thus forming a self-activation loop [43]. It is also possible that the formation of BLR–STM heterodimer during embryogenesis is critical for initiating the *STM* self-activation loop, allowing self-maintenance of the shoot meristem.

Our functional analyses also demonstrate that the *LAS* gene contributes to embryonic shoot meristem formation and cotyledon separation downstream of CUC1 and CUC2. *LAS* encodes a putative transcriptional regulator of the GRAS family and acts as a central hub in the gene regulatory network for axillary meristem formation downstream of CUC2 [23,24]. The precise mechanisms by which *LAS* regulates these processes remain elusive; however, the mutation in the *LAS* ortholog in tomato affects the levels of hormones involved in shoot meristem activity, such as auxin, gibberellin (GA), and cytokinin [44,45]. Recently, it has been shown that the LAS protein binds to the promoter of *GA2ox4*, which encodes a GA deactivation enzyme, and promotes its expression [46]. These results raise the possibility that *LAS* reduces the levels of GA in the boundary region to promote shoot meristem activity, thereby contributing independently of *STM*, *KNAT6*, and *BLR* to the process downstream of the *CUC* genes.

In contrast to the above four genes, which are positive regulators of shoot meristem activity, *KLU/CYP78A5* represents the functionally opposite class of downstream genes regulated by *CUC1* and *CUC2*, as shown by the enhanced shoot meristem size and activity in the *klu* mutant as well as its genetic interaction with the *stm* mutant. Our data thus indicate that, by activating the two classes of genes with opposing functions during embryogenesis, the *CUC* genes create an optimal microenvironment for the shoot meristem to maintain its appropriate size and to produce organs at an appropriate rate. It has been well established that the balance between the self-renewal of stem cells and differentiation of their progeny is critical for postembryonic shoot meristem and that this balance is maintained by the WUS–CLV3 feedback system, which is supported by multiple transcription factors and plant hormones [3,47]. Our results provide an additional level of regulation for the balancing mechanism of shoot stem cell maintenance. Detailed functional analysis of the *KLU* gene as well as its relationship to previously known stem cell regulators will further improve our understanding of shoot meristem regulation.

## 4. Materials and Methods

### 4.1. Plant Materials

*Arabidopsis thaliana* accessions Columbia (Col) and Landsberg *erecta* (L*er*) were used as the wild type. *CUC1-GR* was established in the L*er* background using the same construct as described previously [22]. *CLV3::GUS* was reported previously [33]. Expression analyses of genes downstream of *CUC* were carried out in *cuc1-1 cuc2-1* [4,5]. For rescue experiments, *cuc1-5 cuc2-3* was used [6]. Loss-of-function mutants of the downstream genes were as follows: *klu-019348* (Sallk_019348) and *klu-4* for *KLU* [25]; *knat6-2* for *KNAT6* [19]; *las-101* for *LAS* [6]; *stm-1C* for *STM* [29]; and *pny-40126* for *BLR* [48]. Details of the mutants are described in Appendix A.

### 4.2. Constructs

For *ProCUC2:LAS*, we amplified the *LAS* coding sequence derived from Col with the PCR primers BamHI-LAS-F (5′-TATCTGGATCCATGCTTACTTCCTTCAAATC-3′) and EcoRI-LAS-R (5′-TTCTCGAATTCTCATTTCCACGACGAAACGG-3′) and placed it upstream of the *35S* terminator in a modified UAS cassette vector [49] using the EcoRI and BamHI sites. The BamHI-NotI fragment containing *LAS* and the terminator was then placed downstream of the *CUC2* promoter of *pBS-gC2*, which contains a 5.9 kb fragment of the *CUC2* genomic sequence [50], yielding *ProCUC2:LAS BS*. The SalI-NotI fragment of *ProCUC2:LAS BS* was then inserted into pBIN50, a modified binary vector carrying a kanamycin resistance gene [29]. To obtain *ProCUC2:BLR*, cDNA derived from L*er* was amplified using the primers BLRcDNAfull-F (5′-TTTCCCATGGCTGATGCATA-3′) and BLRcDNAfull-R (5′-TCAACCTACAAAATCATGTA-3′), cloned into pCR™-Blunt II-TOPO (Invitrogen, [Waltham, MA, USA]), and the resulting EcoRI fragment was then placed upstream of the 35S terminator of the modified UAS cassette. Fusion with the *CUC2* promoter and transfer to a binary vector was carried out in the same manner as that for *ProCUC2:LAS*, except that pBIN60, a modified binary vector with a sulfadiazine resistance gene, was used. To obtain the other chimeric constructs of the CUC2 promoter and downstream genes, the 3.1 kb SalI-BglII fragment of the CUC2 promoter was blunt-ended and cloned into the blunt-ended HindIII site of the gateway destination vector pGWB1 carrying kanamycin and hygromycin resistance genes [51], resulting in *ProCUC2 pGWB1*. To generate entry clones, cDNAs were first amplified with gene-specific primers and then with the *attB* adaptor primers listed in Appendix A and cloned into pDONR221 by BP reaction. The inserts were then transferred to *ProCUC2 pGWB1* by LR reaction. For *ProCUC2:GUS*, the *GUS* gene fragment was transferred from the entry clone pENTR-gus (Invitrogen [Waltham, MA, USA]) to *ProCUC2 pGWB1* via LR reaction. Plant transformation was carried out by the floral dip method [52]. The *Pro2kb* and *Pro1kb* reporter constructs of the *KLU* gene contain −2066 to +16 and −1061 to +16 sequences from the start codon, respectively, at their 5′ end of *vYFPer*, an ER-localized version of Venus, and −85 to +338 sequence from the stop codon at their 3′ end and are cloned in the binary vector pBarMAP [53]. Both constructs were transformed to L*er*.

### 4.3. Rescue Experiments

Double-homozygous *cuc1-5 cuc2-3* plants were seedling lethal and did not produce flowers, thus each construct for the rescue experiments was transformed to *cuc1-5 cuc2-3/*+ plants via the floral dip method. T_1_ seeds were selected for drug resistance on Murashige–Skoog plates [54] and the phenotypes of the resistant seedlings were scored. Subsequently, double-homozygous plants were identified by PCR-based genotyping. For analyses of the combined effect of *STM* with *LAS*, one transgenic line containing the *ProCUC2:STM* transgene was maintained and T_2_ plants with the *cuc1-5 cuc2-3/*+ genotype were crossed with two independent lines of *ProCUC2:LAS* with the *cuc1-5 cuc2-3/+* genotype. The seedling phenotype was first scored in the F_2_ generation and the genotype of the *CUC2* locus as well as the presence of each transgene was subsequently analyzed. The same *ProCUC2:STM* line was used for analysis of the combined effect of *STM* with the rest of the genes and was crossed with two independent transgenic lines for each gene. Scoring of the seedling phenotype and subsequent genotyping were carried out in the F_1_ generation. To classify the phenotypes into the mild and strong categories, we examined the presence or absence of visible shoot under a binocular. The absence of shoot production was confirmed at 14 dag or later.

### 4.4. Histological Analysis

In situ hybridization was carried out as described previously [17]. As a template for the *UFO* probe, we used pDW221.1 [55]. For generating templates for other probes, gene-specific fragments were PCR-amplified using the primers listed in Appendix A and cloned into pCR™-Blunt II-TOPO. For GUS detection, embryos were dissected and immediately stained in staining solution with 5 mM ferricyanide and ferrocyanide [56]. Embryos and seedling apices were visualized after clearing as described previously [17].

### 4.5. DEX Induction and qRT-PCR

Induction of CUC1-GR with DEX and expression analysis by qRT-PCR were carried out as described previously [22]. Primers used for qRT-PCR are listed in Appendix A.

## Figures and Tables

**Figure 1 ijms-21-05864-f001:**
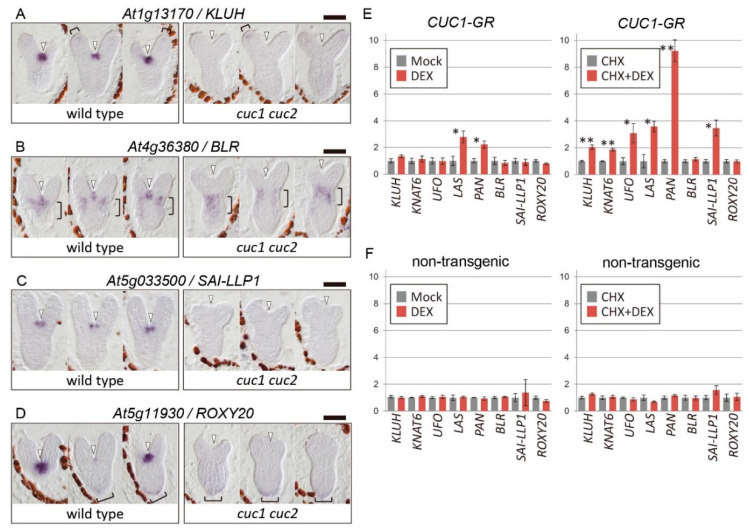
Regulation of candidate downstream genes by *CUC1* and *CUC2*. (**A**–**D**) In situ hybridization of newly identified candidate downstream genes. Four of the six candidates are shown. Three serial longitudinal sections of wild-type L*er* (**left**) and *cuc1-1 cuc2-1* double-mutant (**right**) embryos at the late heart stage. Arrowheads indicate the position of the cotyledon boundary region. Brackets in A, B, and D indicate the position of expression outside the boundary region. Bars = 50 μm. (**E**,**F**) Transcriptional responses of candidate downstream genes upon dexamethasone (DEX) treatment in *CUC1-GR* (**E**) and non-transgenic (**F**) plants in the absence (**left**) or presence (**right**) of the protein synthesis inhibitor cycloheximide (CHX). Three biological replicates of 7-day-old seedlings. Single and double asterisks indicate *p* < 0.05 and *p* < 0.01, respectively, in comparisons between samples with and without DEX (Welch’s *t*-test).

**Figure 2 ijms-21-05864-f002:**
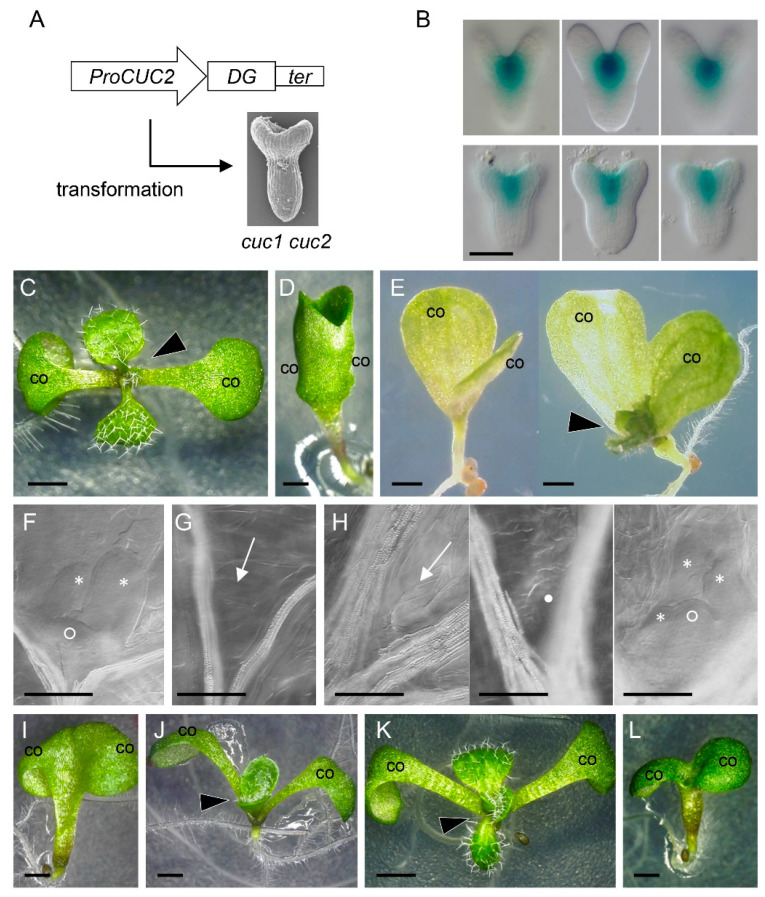
Rescue of *cuc1 cuc2* phenotype by candidate downstream genes. (**A**) Schematic diagram of rescue experiments. *CUC2* promoter (*ProCUC2*), cDNA of downstream gene (*DG*), and nos terminator (*ter*). (**B**) *CUC2* promoter activity detected by β-glucuronidase (GUS). Longitudinal views of GUS-stained wild-type Col (**top**) and *cuc1-5 cuc2-3* (**bottom**) embryos in three different optical sections, showing expression in the boundary region. (**C**,**D**) Seedlings of wild-type Col (**C**) and *cuc1-5 cuc2-3* (**D**), 7 days after germination (dag). Wild type has two separated cotyledons (co) and develops a shoot between them (arrowhead), whereas *cuc1-5 cuc2-3* has cotyledons (co) fused along their margins. (**E**) A *cuc1-5 cuc2-3* seedling mildly rescued by the *STM* transgene at 9 dag (**left**) and 16 dag (**right**). Cotyledons (co) are partially fused on one side. Note that the shoot is only visible at 16 dag (arrowhead). (**F**,**G**) Shoot apices in cleared seedlings (11 dag) of wild type (**F**) and *cuc1-5 cuc2-3* (**G**). Wild type develops the shoot meristem (open circle) and leaf primordia (asterisks), whereas *cuc1-5 cuc2-3* lacks these structures at the corresponding postion (arrow). (**H**) Shoot apices in cleared seedlings (11 dag) of *cuc1- 5 cuc2-3* rescued by *STM*, showing variable phenotypes: no visible shoot meristem and leaf primordia (**left**, arrow), small undifferentiated tissue (**middle**, closed circle), and shoot meristem with leaf primordia (**right**, open circle, and asterisks). (**I**–**L**) *cuc1-5 cuc2-3* seedlings (9 dag) rescued by combined transgenes: *STM* and *LAS* (**I**,**J**), *STM* and *BLR* (**K**), and *STM* and *KNAT6* (**L**). Arrowheads represent emerging shoot. co, cotyledon. Bars = 50 μm (**B**,**F**–**H**); 1 mm (**C**–**E**,**I**–**L**).

**Figure 3 ijms-21-05864-f003:**
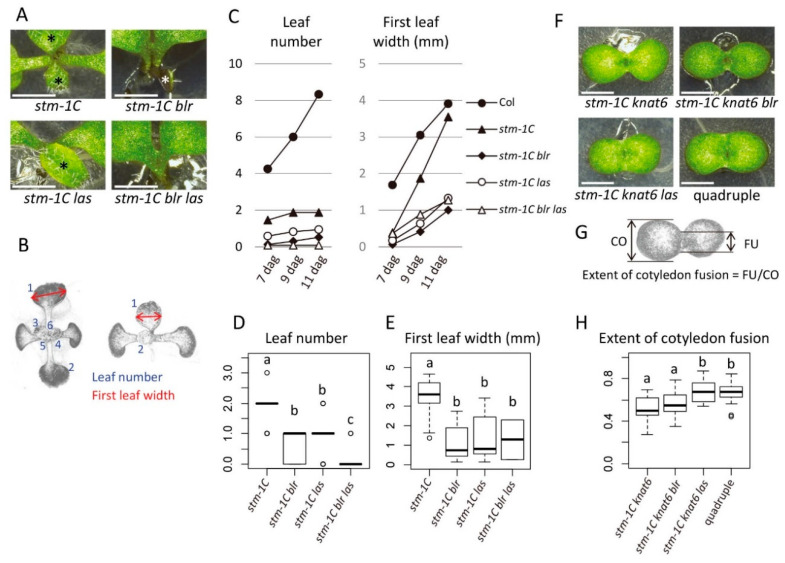
Genetic interactions of *stm, las, blr*, and *knat6* mutants. (**A**) The *las* mutation enhances the shoot production phenotype of *stm-1C* and *stm-1C blr* mutants. Shoot apex at 9 dag. Asterisks indicate developing leaves. (**B**) Examples of shoot phenotype measurements in the wild-type Col (**left**) and *stm-1C* (**right**). (**C**) Change in leaf number (**left**) and first leaf width (**right**). (**D**,**E**) Box plots of leaf number (**D**) and first leaf width (**E**) at 11 dag. (**F**) *las* enhances the cotyledon fusion phenotype of *stm-1C knat6* and *stm-1C knat6 blr* mutants. Seedlings at 7 dag. (**G**) Quantification of cotyledon fusion. (**H**) Box plot showing extent of cotyledon fusion at 7 dag in each genotype. Different letters in box plots indicate statistically significant differences (*p* < 0.01, Steel–Dwass method for **D**; *p* < 0.05, Tukey–Kramer method for **E** and **H**). Sample size is 24, 23, 17, and 23 for **C** (**left**) and **D**; 24, 12, 13, and 2 for **C** (**right**) and **E**; and 11, 23, 12, and 19 for **H**. Bars in **A** and **E**, 2 mm.

**Figure 4 ijms-21-05864-f004:**
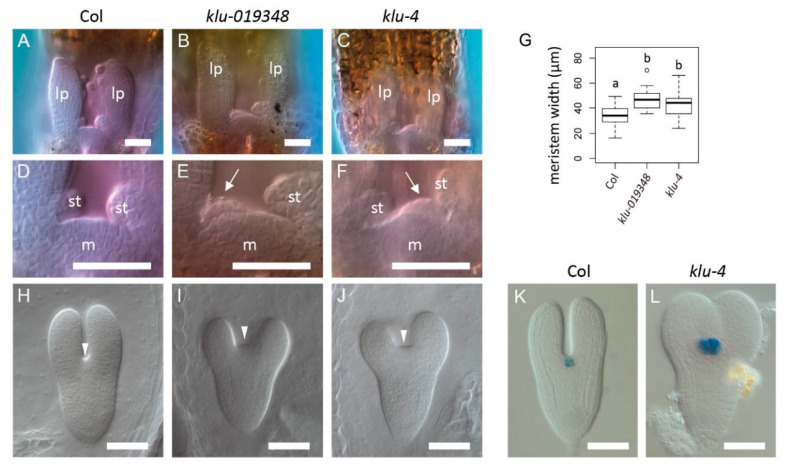
Mutations in *KLU* cause shoot meristem enlargement and precocious leaf formation. (**A**–**F**) Shoot apex of wild type (**A**,**D**), *klu-019348* (**B**,**E**), and *klu-4* (**C**,**F**) in cleared seedlings at 3 dag. (**D**–**F**) are close-up views of (**A**–**C**), respectively. Optical sections are photographed at a position slightly off-center to reveal the precociously formed third-leaf primordia in the *klu* mutants (arrows). lp, the first two leaf primordia; m, shoot meristem; st, stipule. (**G**) Shoot meristem width of wild type and two *klu* mutant alleles at 3 dag. Different letters in box plots indicate statistically significant differences (*p* < 0.01, Tukey–Kramer method). Sample size is 34, 28, and 42 for **G**. (**H**–**J**) Torpedo-stage embryos of wild type (**H**), *klu-019348* (**I**), and *klu-4* (**J**). Arrowheads indicate the position of the cotyledon boundary region. (**K**,**L**) *CLV3::GUS* expression in wild-type (**K**) and *klu-4* (**L**) torpedo-stage embryos. Scale bar = 50 μm.

**Figure 5 ijms-21-05864-f005:**
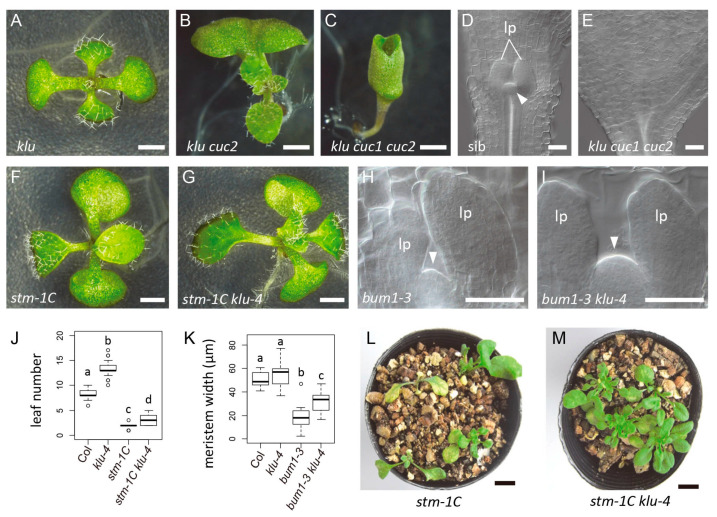
*KLU* acts downstream of *CUC1* and *CUC2*, and counteracts *STM* in the regulation shoot meristem activity. (**A**–**C**) Seven-day-old seedlings of *klu-4* (**A**), *klu-4 cuc2-3* (**B**), and *klu-4 cuc1-5 cuc2-3* (**C**). (**D**,**E**) Shoot apices of the progeny from *klu-4 cuc1-5 cuc2-3*/*+* parent plants. A sib seedling with the shoot meristem (**D**) and a *klu cuc1 cuc2* seedling without it (**E**). (**F**,**G**) Nine-day-old seedlings of the strong *stm* allele *stm-1C* (**F**) and *stm-1C klu-4* (**G**). (**H**,**I**) Shoot apex of the weak *stm* allele *bum1-3* (**H**) and *bum1-3 klu-4* (**I**) at 4 dag. (**J**) Leaf number at 11 dag. (**K**) Shoot meristem width at 4 dag. (**L**,**M**) Twenty-nine-day-old plants of *stm-1C* (**L**) and *stm-1C klu-4* (**M**). Four plants are grown in each pot. Different letters in box plots indicate statistically significant differences (*p* < 0.01, Steel–Dwass method for **J**; *p* < 0.05, Tukey–Kramer method for **K**). Sample size is 55, 30, 24, and 14 for **J**; 14, 18, 15, and 9 for **K**. Arrowheads indicate the shoot meristem. lp, leaf primordia. lp, leaf primordia. Scale bar, 1 mm for **A** to **C**, **F**, and **G**; 50 μm for **D**, **E**, **H**, and **I**; 10 mm for **L** and **M**.

**Figure 6 ijms-21-05864-f006:**
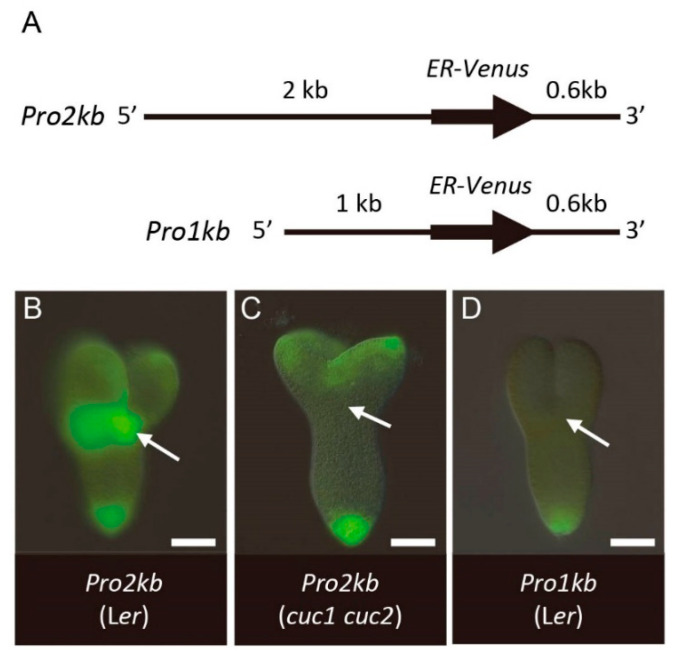
*KLU* expression is regulated by *CUC1* and *CUC2* via a specific promoter region. (**A**) Schematic diagram of the *KLU* reporter genes. (**B**–**D**) Expression of *KLU* reporter genes in embryos. *Pro2kb* in wild type L*er* (**B**), *Pro2kb* in *cuc1-1 cuc2-1* (**C**), and *Pro1kb* in wild type L*er* (**D**). Arrows indicate the position of the cotyledon boundary region of wild type (**B**,**D**) and the corresponding region of *cuc1-1 cuc2-1* (**C**). Scale bar, 50 μm.

**Table 1 ijms-21-05864-t001:** Rescue of *cuc1 cuc2* seedling phenotype by downstream gene expression under the control of *CUC2* regulatory sequence.

Transgene	No Rescue ^a^ (%)	Mild Rescue ^b^ (%)	Strong Rescue ^c^ (%)	Total Number of T1 Seedlings
*KLU*	100	0	0	25
*KNAT6*	100	0	0	15
*UFO*	100	0	0	20 ^d^
*LAS*	86.5	13.5	0	37
*STM*	54.5	45.5	0	11
*PAN*	100	0	0	12
*LSH4*	100	0	0	31
*BLR*	100	0	0	15
*SAI-LLP1*	100	0	0	12
*ROXY20*	100	0	0	12
*CUC2*	8.3	50.0	41.7	12
*GUS*	100	0	0	4

^a^ Cotyledons were fused along both sides with only a small split at their tips and no shoot was formed. ^b^ Cotyledons were fused along one or both sides with more than half of their margins split. No shoot was formed or a shoot became visible only after 9 dag. ^c^ Cotyledons were completely separated and a shoot was visible by 9 dag. ^d^ One seedling had flat and round green tissue on top of the hypocotyl instead of a cup-shaped cotyledon. This is classified as “No rescue”, as neither shoot formation nor cotyledon separation occurred.

**Table 2 ijms-21-05864-t002:** Effect of combined expression of *STM* and *LAS* on the phenotype of *cuc1 cuc2.*

Transgene A	Transgene B	No Rescue ^a^ (%)	Mild Rescue ^b^ (%)	Strong Rescue ^c^ (%)	Total Number of F2 Seedlings	Group ^d^
*STM*	-	100	0	0	11	a
-	*LAS*	100	0	0	26	a
*STM*	*LAS*	0	71.1	28.9	38	b

^a^ Cotyledons were fused along both sides with only a small split at their tips and no shoot was formed. ^b^ Cotyledons were fused along one or both sides with more than half of their margins split. No shoot was formed or a shoot became visible only after 9 dag. ^c^ Cotyledons were completely separated and a shoot was visible by 9 dag. ^d^ Different letters indicate statistically significant differences (*p* < 0.01, Fisher’s exact test with Holm multiple testing correction).

**Table 3 ijms-21-05864-t003:** Effect of combined expression of *STM* and other downstream genes on the phenotype of *cuc1 cuc2*.

Transgene A	Transgene B	No Rescue ^a^ (%)	Mild Rescue ^b^ (%)	Strong Rescue ^c^ (%)	Total Number of F1 Seedlings
*STM*	*−*	92.3	7.7	0	13
*−*	*KNAT6*	100	0	0	6
*STM*	*KNAT6*	42.9	57.1	0	14*
*STM*	*−*	100	0	0	5
*-*	*UFO*	100	0	0	14
*STM*	*UFO*	100	0	0	12
*STM*	*−*	100	0	0	4
*−*	*PAN*	100	0	0	5
*STM*	*PAN*	100	0	0	9
*STM*	*−*	93.8	6.3	0	32
*−*	*LSH4*	100	0	0	19
*STM*	*LSH4*	92.9	7.1	0	14
*STM*	*−*	100	0	0	7
*−*	*BLR*	100	0	0	3
*STM*	*BLR*	16.7	0	83.3	6 **
*STM*	*−*	100	0	0	10
*−*	*GUS*	100	0	0	9
*STM*	*GUS*	100	0	0	10

^a^ Cotyledons were fused along both sides with only a small split at their tips and no shoot was formed. ^b^ Cotyledons were fused along one or both sides with more than half of their margins split. No shoot was formed or a shoot became visible only after 9 dag. ^c^ Cotyledons were completely separated and a shoot became visible by 9 dag. Asterisks indicate significant differences in the ratio of the phenotypes of plants carrying the transgene B alone and those carrying both transgenes A and B (* *p* < 0.05, ** *p* < 0.01; Fisher’s exact test).

**Table 4 ijms-21-05864-t004:** Frequency of cotyledon fusion in *stm* mutant combined with *blr* and *las* mutants.

Genotype	Frequency of Fusion (%)	Total	Group *
Col	0	55	a
*stm*	12.5	24	b
*blr stm*	91.3	23	c
*las stm*	35.3	17	b
*blr las stm*	95.7	23	c

* Different letters indicate statistically significant differences (*p* < 0.01, Fisher’s exact test with Holm multiple testing correction).

**Table 5 ijms-21-05864-t005:** Genetic interactions among *klu-4*, *cuc1-5*, and *cuc2-3*.

Genotype	Phenotype			Total Number of Seedlings
	Normal ^a^ (%)	Weak ^b^ (%)	Strong ^c^ (%)	
Col	100	0	0	76
*klu*	100 *	0	0	289
*cuc1*	95.6	4.4	0	135
*cuc2*	100	0	0	118
*klu cuc1*	95.9	4.1	0	172
*klu cuc2*	97.2	2.8	0	217
*cuc1 cuc2*	0	0	100	50
*klu cuc1 cuc2*	0	0	100	54

^a^ Cotyledons were completely separated and a shoot was formed immediately after germination. ^b^ Cotyledons were partially fused and a shoot was formed immediately after germination. ^c^ Cotyledons were strongly fused, forming a cup shape, and no shoot was formed after two weeks of observation. * A very small fraction (1.7%) had three cotyledons instead of two.

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
