# Peer review of "Establishment of the Embryonic Shoot Meristem Involves Activation of Two Classes of Genes with Opposing Functions for Meristem Activities"

_ijms, 2020, doi:10.3390/ijms21165864_

Round 1
Reviewer 1 Report
I have confirmed that the manuscript has been revised adequately in careful consideration of the reviewers’ comments.
Reviewer 2 Report
The authors have now addressed the majority of my concerns.
This manuscript is a resubmission of an earlier submission. The following is a list of the peer review reports and author responses from that submission.
Round 1
Reviewer 1 Report
This manuscript addresses genetic pathways that contribute to embryonic meristem formation in Arabidopsis.
The authors focus on the CUC1 and CUC2 genes and aim to identify downstream targets that contribute directly to meristem formation. Many of the genes they identify have previously been assigned roles in shoot meristem or leaf development.
The manuscript is well written and the figures are generally of high quality. The study builds on previous results from the group. However, it does seem to read more like an aggregation of unrelated results rather than a focussed study.
I believe the main result is that KLU is possibly a direct target activated by CUC1/2, although this is counter intuitive given that KLU is normally repressing shoot meristem development (in addition to leaf and ovule development), while CUC1/2 are promoting the same process. This relationship is not really addressed in detail, although it is suggested to counteract STM activity. It is challenging to rationalise how all of the data fits together in a functional model.
General Comments:
- The authors suggest the presence of a “CHX-sensitive negative” factor that limits induction of targets by DEX alone. This is not discussed in detail – is it really CHX or a limitation of the system? Does the indirect activation of targets flood the transcriptional machinery such that the activation of direct targets is limited? Is the response dose-dependent? I am concerned that this is the only assay used to "confirm" direct targets.
- There is little consistency between time points when measuring phenotypes. For example, Figure 2 shows seedlings at 7, 9 or 11 dag. No reason is provided for this variation.
- Most tables appear to show “number of seedlings” analysed rather than “percentages of seedlings” showing phenotypes. In general, the number of seedlings analysed in these tables appears insufficient for robust statistical analysis (e.g. depending on the tables, the number varies from 1 to 67 seedlings analysed). This needs to be corrected.
- It is unclear what the pKLU experiments add, given that the in situ result is already shown in Figure 1. Also, the in situ for KLU and the promoter pattern appear different (Figure 1 vs Figure 6). Tis needs to be explained.
- The authors use the 35S:CUC1-GR system to assess whether genes showing deregulated expression in CUC1 and/or CUC2 embryos could be direct targets. As per previous studies, they use the 35S promoter and 7 day-old seedlings for the inducible system. This system is quite outdated and presents challenges when it comes to verifying direct targets in the context of the embryonic meristem. More compelling evidence would come from a construct with the endogenous promoter (pCUC1-CUC1-GR) or a ChIP assay using a functional pCUC1:CUC1-GFP.
- Line 46 – in terms of positional information, I wonder why the authors do not consider the mir394/WUS pathway that appears to provide apical/basal information to help establish the meristem.
- Figure 1 – I suggest labelling the “WT” and “cuc1/2” panels
Response
- This manuscript addresses genetic pathways that contribute to embryonic meristem formation in Arabidopsis. The authors focus on the CUC1 and CUC2 genes and aim to identify downstream targets that contribute directly to meristem formation. Many of the genes they identify have previously been assigned roles in shoot meristem or leaf development. The manuscript is well written and the figures are generally of high quality. The study builds on previous results from the group. However, it does seem to read more like an aggregation of unrelated results rather than a focussed study. I believe the main result is that KLU is possibly a direct target activated by CUC1/2, although this is counter intuitive given that KLU is normally repressing shoot meristem development (in addition to leaf and ovule development), while CUC1/2 are promoting the same process. This relationship is not really addressed in detail, although it is suggested to counteract STM activity. It is challenging to rationalise how all of the data fits together in a functional model.
(Response) Thank you for the critical and productive comments. As described in the title and the abstract, our main conclusion is that CUC1 and CUC2 are required for activation of both positive and negative factors for meristem activity. We agree that KLU being downstream of CUC1 and CUC2 is a main finding, but at the same time, we would put equivalent importance to our genetic experiments showing that STM, KNAT6, BLR, and LAS are functionally important downstream regulators in CUC1 and CUC2. This conclusion is not trivial, considering that many other downstream genes related to meristem functions did not rescue the cuc1 cuc2 phenotype even in combination with STM (Table 1 and Table 3) and that stm knat6 blr las quadruple mutant phenocopies cuc1 cuc2 double mutant in that they lack shoot meristem and form strongly fused cotyledons (Figure 3). So we consider that providing evidence for both positive and negative factors in a single paper is not mere aggregation of unrelated results.
We also agree that identification of KLU as a CUC downstream gene is counter intuitive. However, this conclusion is strongly supported by at least three pieces of evidence, in situ hybridization, reporter analysis, and CUC1-GR experiments (Figure 1 and Figure 6). Although KLU expression is detected additionally in regions outside the CUC gene expression domain, the specific loss of KLU expression in the boundary region of cuc1 cuc2 mutant strongly support the specific regulatory roles for the CUC genes in KLU expression.
To address the relationship between KLU and CUC in more detail, we newly added data of genetic interaction among these genes (Table 5, Figure 5A to 5E, Line 277 to 287). Importantly, the phenotype of the klu cuc1 cuc2 triple mutant is essentially the same as that of cuc1 cuc2, showing that the effect of the klu mutation on shoot meristem activity is completely dependent of the CUC1 and CUC2 genes. This clear epistatic relationship contrasts with the relationship between KLU and STM, where the effect of their mutations counteracts each other. Together, these results indicate that CUC1 and CUC2 are upstream of both STM and KLU and meristem activity is balanced by the interaction of the two downstream genes. We hope that this conclusion rationalizes our results.
General Comments:
- The authors suggest the presence of a “CHX-sensitive negative” factor that limits induction of targets by DEX alone. This is not discussed in detail – is it really CHX or a limitation of the system? Does the indirect activation of targets flood the transcriptional machinery such that the activation of direct targets is limited? Is the response dose-dependent? I am concerned that this is the only assay used to "confirm" direct targets.
(Response) We understand that the results of CUC1-GR provide indicative but not conclusive evidence for direct targets. We therefore changed the text so that we avoided concluding that the six genes activated in the presence of CHX were direct targets (Line 97 to 98). We also modified the abstract (Line 23 to 32) and the final part of introduction (Line 79) to avoid any statement about transcriptional regulation. Regarding the possibility of indirect activation of targets that inhibit the activation of CUC1 direct target genes, we added a sentence that point out such possibility (Line 109 to 111).
- There is little consistency between time points when measuring phenotypes. For example, Figure 2 shows seedlings at 7, 9 or 11 dag. No reason is provided for this variation.
(Response) Thank you for pointing this out. For Figure 2, we added 9 dag seedling of cuc1 cuc2 rescued by ProCUC2:STM (Figure 2E, left panel) so that readers can compare the phenotype with plants rescued by two transgenes at the same age (Figures 2I to L). 9 dag is the timing where we judged the absence/presence of shoot formation, as described in the notes of Table 1, 2, and 3. In addition, we removed the SEM image of 7 dag (Figure 2I of previous version), because it was not very informative, and instead added images of cleared seedling apices to show variability of the phenotype in rescued plants (Figure 2H). The age of plants in these panels is all 11 dag and is the same as that of Figures 2F and 2G. We also noticed that the previous main text and figure legend did not fully explained the results, so added a more complete description in the new version (Line 131 to 147, Line 157 to 172, Line 438 to 440). In addition, we noticed that the age of wild type and cuc1 cuc2 mutant plants in Figures 2C and 2D was wrong and it was 7 dag instead of 9 dag. We apologize for this error.
- Most tables appear to show “number of seedlings” analysed rather than “percentages of seedlings” showing phenotypes. In general, the number of seedlings analysed in these tables appears insufficient for robust statistical analysis (e.g. depending on the tables, the number varies from 1 to 67 seedlings analysed). This needs to be corrected.
(Response) Now we unified the style of the tables so that they show “percentages of seedlings” for each category together with the total number. We also added statistical analyses for the data presented in Table 2 and 3 and these analyses supported our conclusions regarding the importance of combined activities between STM and LAS, BLR, and KNAT6. Please also see our response to the fourth comment of reviewer 2.
- It is unclear what the pKLU experiments add, given that the in situ result is already shown in Figure 1. Also, the in situ for KLU and the promoter pattern appear different (Figure 1 vs Figure 6). This needs to be explained.
(Response) We consider that the pKLU reporter experiments give an additional piece of evidence for the dependence of KLU expression on CUC1 and CUC2 activity. Given the functional importance of this gene as a CUC downstream gene, it would be worth providing independent evidence with a different method. The difference between KLU expression patterns in in situ hybridization and in reporter analysis was added to the text (Line 320 to 324).
- The authors use the 35S:CUC1-GR system to assess whether genes showing deregulated expression in CUC1 and/or CUC2 embryos could be direct targets. As per previous studies, they use the 35S promoter and 7 day-old seedlings for the inducible system. This system is quite outdated and presents challenges when it comes to verifying direct targets in the context of the embryonic meristem. More compelling evidence would come from a construct with the endogenous promoter (pCUC1-CUC1-GR) or a ChIP assay using a functional pCUC1:CUC1-GFP.
(Response) As we discussed in the response to the first general comment, we realize that the CUC1-GR experiments (the promoter is not 35S but RPS5A) alone do not give conclusive evidence for direct targets, so we avoid concluding so in the revised version. Nevertheless, we consider that ectopic and forced expression experiments can still be an informative method to examine the ability of regulatory genes to affect target gene expression directly or indirectly.
- Line 46 – in terms of positional information, I wonder why the authors do not consider the mir394/WUS pathway that appears to provide apical/basal information to help establish the meristem.
(Response) Here we do not intend to discuss positional information as a main problem, but rather consider the process that coordinate expression of regulatory genes required for shoot meristem activity during embryogenesis at the cotyledon boundary region. We therefore removed “at an appropriate position” from the sentence (Line 49). However, we agree that the regulation of WUS activity by miR394 and LCR is another important pathway for shoot meristem initiation, so cited this paper (Line 51, reference #11).
- Figure 1 – I suggest labelling the “WT” and “cuc1/2” panels
(Response) Thank you for the suggestion for improving the figure. We added the labels accordingly (wild type vs cuc1 cuc2 in Figure 1A–D; CUC1-GR vs non-transgenic in Figure 1E and F; wild type vs cuc1 cuc2 in Figure S1B).
Reviewer 2 Report
In the study described in this paper, downstream genes of CUC1 and CUC2 involved in embryonic shoot meristem establishment were investigated by several lines of expression analyses and functional analyses based on mutant phenotypes and rescue experiments, which identified five important downstream genes: STM, LAS, BLR, KNAT6, and KLU. Of these, LAS, BLR, and KNAT6 were found to act together with STM in the positive regulation of shoot meristem and boundary formation, while KLU was shown to regulate negatively shoot meristem in dependence on CUCs. This finding greatly contributes to our understanding of the gene regulatory network underlying embryonic shoot meristem establishment.
The manuscript is well written and the conclusion sounds reasonable. I have no serious criticisms. The following are just minor specific comments.
- The format of tables is partially broken probably due to the problem of conversion of the original file into PDF.
- Table 1 shows the total number of samples and percentages of categories while Tables 2 and 3 show absolute numbers of categorized samples. Table 4 shows absolute numbers of categorized samples along with percentages of the specific category. It may be more reader-friendly to unify the style of these tables.
- I should like to suggest that the titles of Table 2 and Table 3 are changed to include information about the genetic background: for example, “Effect of combined expression of … on the phenotype of cuc1 cuc2”.
- It is desirable to perform some appropriate statistical test for the data presented in Tables 1, 2, and 3.
- At the leftmost column of Figure 3H, “stm-1 kna6” should read “stm-1 knat6”.
Response
Comments and Suggestions for Authors
In the study described in this paper, downstream genes of CUC1 and CUC2 involved in embryonic shoot meristem establishment were investigated by several lines of expression analyses and functional analyses based on mutant phenotypes and rescue experiments, which identified five important downstream genes: STM, LAS, BLR, KNAT6, and KLU. Of these, LAS, BLR, and KNAT6 were found to act together with STM in the positive regulation of shoot meristem and boundary formation, while KLU was shown to regulate negatively shoot meristem in dependence on CUCs. This finding greatly contributes to our understanding of the gene regulatory network underlying embryonic shoot meristem establishment. The manuscript is well written and the conclusion sounds reasonable. I have no serious criticisms. The following are just minor specific comments.
- The format of tables is partially broken probably due to the problem of conversion of the original file into PDF.
(Response) Thank you for pointing this out. This happened during the process of conversion after we submitted the original file. We have reformatted the tables.
- Table 1 shows the total number of samples and percentages of categories while Tables 2 and 3 show absolute numbers of categorized samples. Table 4 shows absolute numbers of categorized samples along with percentages of the specific category. It may be more reader-friendly to unify the style of these tables.
(Response) Thank you very much for the valuable suggestion. Now we show percentages of different categories and the total number of observed seedlings in all tables.
- I should like to suggest that the titles of Table 2 and Table 3 are changed to include information about the genetic background: for example, “Effect of combined expression of … on the phenotype of cuc1 cuc2”.
(Response) We changed the title accordingly.
- It is desirable to perform some appropriate statistical test for the data presented in Tables 1, 2, and 3.
(Response) We carried out statistical analysis for the data in Tables 2 and 3 but not for those in Table 1 (see later). In Table 2, we tested the differences in the observed ratios of the phenotypes between STM alone, LAS alone, and their combination. The differences were statistically significant.
For the data in Table 3, we tested the effect of the transgenes B (KNAT6, UFO, PAN, etc) on the phenotype of cuc1 cuc2 plants carrying the STM transgene. To carry out this statistical test, we had to reorganize the structure of the table. As described in materials and methods, the data in this table were obtained by crossing cuc1 cuc2/+ plants carrying the STM transgene with those carrying a transgene B and scoring the phenotypes of all F1 plants, which were subsequently subjected to PCR analysis to determine their genotypes as well as the presence/absence of each transgene. In the previous version of Table 3, we presented the merged number of cuc1 cuc2 plants carrying STM alone from all crosses for simplicity (the 2nd row, total 71 plants (67 + 4 + 0)), but now in the revised version, we divided it into those obtained from crosses involving different transgenes B, so that we can carry out the statistical test. The results showed that addition of KNAT6 and BLR significantly increased the frequency of rescued phenotype.
Regarding Table1, we would like to keep the data without statistical tests. This is because the primary purpose of this experiment is to test the ability of each downstream gene to rescue the cuc1 cuc2 mutant phenotype but not to compare their ability one another. We examined at least twelve independent T1 transformants for each downstream gene: we believe that this number is comparable to that of transformants in other published papers by the Arabidopsis community when they examine the effect of a transgene. In case of STM and LAS, where we observed positive results, we obtained five rescued plants with a similar phenotype for each transgene, showing that the effects of these transgenes were reproducible. Please also note that we screened on average four times as many seedlings presented in Table 1 to obtain transformants that were homozygous for the cuc2 mutation (see materials and methods).
- At the leftmost column of Figure 3H, “stm-1 kna6” should read “stm-1 knat6”.
(Response) Thank you for letting us know this error. We corrected it accordingly